# Providers' mediating role for medication adherence among cancer survivors

**Justin G. Trogdon**[1,2]*, **Krutika Amin**[1¤], **Parul Gupta**[2], **Benjamin Y. Urick**[3], **Katherine E. Reeder-Hayes**[2,4], **Joel F. Farley**[5], **Stephanie B. Wheeler**[1,2], **Lisa Spees**[1,2], **Jennifer L. Lund**[2,6]

**1** Department of Health Policy and Management, Gillings School of Global Public Health, University of North Carolina at Chapel Hill, Chapel Hill, North Carolina, United States of America, **2** Lineberger Comprehensive Cancer Center, University of North Carolina at Chapel Hill, Chapel Hill, North Carolina, United States of America, **3** Division of Practice Advancement and Clinical Education, Eshelman School of Pharmacy, University of North Carolina at Chapel Hill, Chapel Hill, North Carolina, United States of America, **4** Division of Hematology/Oncology, Department of Medicine, University of North Carolina at Chapel Hill, Chapel Hill, North Carolina, United States of America, **5** Department of Pharmaceutical Care and Health Systems, University of Minnesota, Minneapolis, Minnesota, United States of America, **6** Department of Epidemiology, Gillings School of Global Public Health, University of North Carolina at Chapel Hill, Chapel Hill, North Carolina, United States of America

¤ Current address: Kaiser Family Foundation, San Francisco, California, United States of America
* justintrogdon@unc.edu

**Data Availability Statement:** The SEER-Medicare data is owned by the SEER registry Principal Investigators and the Centers for Medicare and Medicaid Services. Although personal identifiers

## Abstract

### Background

We conducted a mediation analysis of the provider team's role in changes to chronic condition medication adherence among cancer survivors.

### Methods

We used a retrospective, longitudinal cohort design following Medicare beneficiaries from 18-months before through 24-months following cancer diagnosis. We included beneficiaries aged ≥66 years newly diagnosed with breast, colorectal, lung or prostate cancer and using medication for non-insulin anti-diabetics, statins, and/or anti-hypertensives and similar individuals without cancer from Surveillance, Epidemiology, and End Results-Medicare data, 2008–2014. Chronic condition medication adherence was defined as a proportion of days covered ≥ 80%. Provider team structure was measured using two factors capturing the number of providers seen and the historical amount of patient sharing among providers. Linear regressions relying on within-survivor variation were run separately for each cancer site, chronic condition, and follow-up period.

### Results

The number of providers and patient sharing among providers increased after cancer diagnosis relative to the non-cancer control group. Changes in provider team complexity explained only small changes in medication adherence. Provider team effects were statistically insignificant in 13 of 17 analytic samples with significant changes in adherence. Statistically significant provider team effects were small in magnitude (<0.5 percentage points).

for all patient and medical care providers have been removed from the SEER-Medicare data, there remains the remote risk of re-identification (given the large amount of data available). Data can be accessed, subject to approval and data use agreement, from the Healthcare Delivery Research Program at the National Cancer Institute (http://appliedresearch.cancer.gov/seermedicare/obtain/requests.html).

**Funding:** This research was support by the National Institute on Aging (NIA R01 AG050733; PI: Trogdon; https://www.nia.nih.gov/). The database infrastructure used for this project was supported through the University of North Carolina Clinical and Translational Science Award (UL1TR001111) and the UNC Lineberger Comprehensive Cancer Center, University Cancer Research Fund via the State of North Carolina. The funders had no role in study design, data collection and analysis, decision to publish, or preparation of the manuscript.

**Competing interests:** Dr. Lund's spouse is a full-time, paid employee of GlaxoSmithKline who also holds stock in the amount of approximately $42,000. Dr. Lund also receives unrelated grant funding paid to her institution from AbbVie. Drs. Reeder-Hayes and Wheeler receive unrelated grant funding paid to their institution from Pfizer. Dr. Farley received unrelated grant funding paid to his institution from Astra Zeneca. All other co-authors have no potential conflicts of interest to report. This does not alter our adherence to PLOS ONE policies on sharing data and materials. The SEER-Medicare data is owned by the SEER registry Principal Investigators and the Centers for Medicare and Medicaid Services. Although personal identifiers for all patient and medical care providers have been removed from the SEER-Medicare data, there remains the remote risk of re-identification (given the large amount of data available). Data can be accessed, subject to approval and data use agreement, from the Healthcare Delivery Research Program at the National Cancer Institute. This does not alter our adherence to PLOS ONE policies on sharing data and materials.

## Conclusions

Increased complexity in the provider team associated with cancer diagnosis did not lead to meaningful reductions in medication adherence. Interventions aimed at improving chronic condition medication adherence should be targeted based on the type of cancer and chronic condition and focus on other provider, systemic, or patient factors.

## Introduction

More than 60% of Medicare beneficiaries diagnosed with cancer also have three or more chronic conditions [1]. Management of chronic conditions among cancer survivors is complex [2–5], especially medication management [6]. Increasing evidence suggests that adherence to medications for chronic conditions decreases in older adults with some cancers [7–13]. For example, our earlier study found that adherence to anti-diabetics and statins declined among older adults with colorectal and lung cancer by two to four percentage points after cancer diagnosis relative to matched non-cancer patients [13].

However, little is known about the *mechanisms* for these changes in chronic condition medication adherence among cancer survivors. A diagnosis of cancer can directly affect medication adherence for other chronic conditions in a variety of ways. Cancer diagnosis can shift the emphasis of medical care to the emerging cancer. For example, in the presence of additional cost and complexity created by cancer-related prescriptions (both treatment and symptom management), patients may decrease adherence to medications for other chronic conditions. Conversely, cancer diagnosis may reinforce the importance of chronic disease prevention, serving as a "wake-up call" and encouraging healthy behaviors such as adherence to medications.

Another pathway for changes in adherence is through changes to the survivor's provider team. After diagnosis, patients see new oncology specialists and may or may not continue to see their original primary care provider and chronic disease specialists even after cancer treatment, which can disrupt communication and coordination among the provider team [4, 5, 14]. Additional providers and medical visits may lead to differential clinical priorities or management strategies across multiple providers, confuse patient-provider communication, or make care coordination more difficult, which may influence chronic condition medication adherence [14].

Understanding the role of the provider team can help inform policy and practice aimed at improving care for patients with multiple chronic conditions. Alternative payment models increasingly put providers at financial risk for holistic patient outcomes, increasing the incentive to coordinate care for chronic conditions. Several quality improvement efforts to improve care for cancer survivors focus on transitioning care from oncology specialists back to primary care providers [15] and improving coordination of care among the provider team [14, 16–18].

In this study, we investigated the mechanisms for cancer-related changes in chronic condition medication adherence among older cancer survivors. Specifically, we conducted a mediation analysis to investigate the role of changes in the provider team structure in adherence changes during and after primary cancer treatment. We hypothesized that the increase in the number of providers and specialists and more complicated patient-sharing relationships will make coordination more difficult and lower medication adherence for chronic conditions [14, 16, 19].

## Methods

### Data source and study populations

We used the linked Surveillance, Epidemiology, and End Results program (SEER) cancer registries and Medicare enrollment and claims data [20]. The SEER registries collect demographic, tumor, and vital status data for incident cancers, covering approximately 34% of the United States population. Medicare enrollment and claims data record longitudinal information about healthcare utilization for beneficiaries enrolled in the fee-for-service program.

We identified patients aged ≥66 with a first primary diagnosis of stage I-III breast, prostate, non-small cell lung, or colorectal cancer from July 1, 2008 –December 31, 2012. Access to the data was originally granted in 2017. We excluded stage IV and metastatic disease for all cancer sites due to the different incentives patients and providers would face for chronic condition medication adherence considering limited life expectancy. Individuals diagnosed at autopsy or death were excluded. Individuals had to have Medicare Parts A, B, and D coverage for the 18-months before through 24-months following the month of cancer diagnosis.

For each condition (i.e., hyperlipidemia, hypertension, and diabetes mellitus), we constructed separate cohorts with at least one International Classification of Diseases, 9th Edition, Clinical Modification diagnosis code for the condition of interest and at least one prescription drug claim for an oral medication to manage that condition from -18 months to -7 months before cancer diagnosis. This approach resulted in 12 distinct cohorts; survivors could be represented in multiple cohorts.

For each cancer-chronic condition cohort, we identified a non-cancer comparison cohort using a 5% random sample of Medicare beneficiaries identified within each SEER region. For each cancer survivor (in each cancer-chronic condition cohort), we identified all potential individuals without a diagnosis of cancer who met the same chronic condition criteria as the cancer-chronic condition cohorts. We matched cancer and non-cancer individuals with the same chronic condition on exact age (in years), sex, race (White, Black, Asian, Hispanic, Native American Indian, Other), and SEER region. Among all eligible individuals, one non-cancer comparison patient was selected at random with replacement and assigned an index date, based on their matched cancer survivor's diagnosis date. As in the cancer cohorts, controls could be in multiple condition cohorts [13].

### Medication adherence

The primary outcome was chronic condition medication adherence, measured using the proportion of days covered (PDC) [21]. The PDC is the number of days covered by a prescription drug divided by the total number of days in an observation window. PDC has been shown to be more reliable than self-report [22, 23] and correlated with drug levels [24]. We removed hospitalizations and skilled nursing facility stays from the denominator and carrying forward any days' supply which overlapped with a hospital or skilled nursing facility stay [21]. Adherence was evaluated at the condition- (e.g., hypertension) level; switching within and across drug classes was allowed.

The PDC was measured in 6-month time windows ending 1) at cancer diagnosis, 2) one year post-diagnosis (after which primary treatment is likely complete [25, 26]), and 3) two years post diagnosis. For the analysis of antidiabetics, we excluded all survivors that initiated insulin at any point during follow-up. PDC calculations are unreliable for insulin, and removal of survivors who initiate insulin is consistent with CMS specifications [27]. The PDC was dichotomized at ≥80% (adherent) versus <80% (non-adherent), a common cut-point [28–37]. As a secondary outcome, we defined discontinuation as a dichotomous indicator equal to

one if the survivor did not fill a drug for their chronic condition for 90 continuous days [38, 39].

## Provider team structure

We defined four measures of the provider team structure. We counted 1) the total number of providers and 2) number of specialists seen by each person in the period. All providers *except* the following were considered specialists: internal medicine doctors without subspecialty training (National Provider Identifier = 207R00000X), family practitioners (207Q00000X), general practitioners (208D00000X), obstetrics and gynecologists (207V00000X, 207VG0400X), and geriatricians (207RG0300X, 207QG0300X). For providers with more than one specialty, they were assigned to the specialty listed in most of their claims. We excluded mid-level providers (e.g., nurse practitioners) due to the variety of regulations on prescriptive authority across states. These measures captured the number of providers and specialists required to coordinate; more providers/specialists make care coordination more difficult [16].

Third, we defined degree as the count of all providers that share patients with the patient's main provider [19]. We designated the main provider using the plurality provider algorithm [40, 41]. This provider team measure captured the level of coordination required between the main provider and all other providers; more provider peers make care coordination more difficult [19].

Fourth, for each pair of providers on a patient's provider team, we calculated the proportion of each providers' patients who were shared with the other provider in the dyad in the previous (6-month) period [14]. We counted a shared patient if two providers billed for outpatient evaluation and management visits. We then calculated the shared patient volume for a provider team using the geometric mean of all pairwise proportions. Shared patient volume was undefined for patients with only one provider; we set shared patient volume to zero in these cases. Shared patient volume captured the recent history of opportunities to coordinate for a patient's provider team; higher levels of shared patient volume represent potentially higher degrees of care coordination.

## Statistical analysis

Our causal model is presented in S1 Appendix. The following analyses were conducted separately in 24 analytic samples defined by combinations of cancer site (n = 4), chronic condition (n = 3), and time comparison (n = 2; diagnosis vs one year post-diagnosis and diagnosis vs two years post-diagnosis). We condensed the four provider team measures to two using factor analysis. We transformed the four provider team measures to be deviations from individual means by subtracting each patient's mean over time from each variable (i.e., $X_{it}^{new} = X_{it} - \bar{X}_{i.}$). Factor analysis using the transformed provider measures consistently identified two factors (with positive eigenvalues) across all analysis samples. The first, "number of providers," had high factor loadings for the number of total providers and number of specialists. The second factor, "sharing among providers," had high factor loadings for degree and shared patient volume. We generated predicted values for these two factors and standardized each factor to be mean zero and in units of standard deviations.

For each analytic sample, we estimated three linear regressions. All regression variables were expressed as deviations from individual means over time to eliminate time-invariant patient characteristics (e.g., tumor features at diagnosis). The first two regressions had the provider team factors as dependent variables and cancer status as the explanatory variable. The third regression had adherence as the dependent variable with cancer and both provider team

factors as explanatory variables. We used seemingly unrelated regression with robust standard errors to stack the within-person regressions into one variance-covariance matrix.

We report the following effects of cancer and provider team on adherence. The natural direct effect (NDE) is the expected difference in adherence between those with and without cancer holding the care team constant in the non-cancer configuration. The natural indirect effect (NIE) is the expected difference in adherence among cancer survivors comparing care teams with and without cancer. Total effect (TE) is the sum of NDE and NIE. See S1 Appendix for derivations.

All statistical analyses were performed using Stata version 15.1 (College Station, TX). This study was approved by the University of North Carolina at Chapel Hill Institutional Review Board.

## Results

The analytic cohort has been described previously [13] and includes 11,831 unique individuals diagnosed with breast cancer, 6,580 with colorectal cancer, 4,105 with lung cancer and 11,879 men diagnosed with prostate cancer, each matched to a non-cancer control. Each cancer cohort experienced notable changes to their provider team after cancer diagnosis compared to their matched non-cancer control group (Table 1). For each cancer cohort, the number of providers and number of specialists increased one-year post-diagnosis relative to the non-cancer control group, which experienced smaller increases over time as the cohort aged. Similarly, each cancer cohort experienced a large increase in their main provider's degree (number of other providers with whom they share patients) in the first year relative to the non-cancer controls. Patient's main provider's shared patient volume, although low overall (i.e., only about 1–2% of patients were shared on average amongst provider team), also increased in the first year for cancer survivors relative to non-cancer controls. The increase in patient volume shared among cancer patients is due to higher rates of patient sharing among oncologists, who often become cancer patients' main provider. For all cancer cohorts and provider team variables, the averages decreased from the first year to the second year post-diagnosis for the cancer cohorts but remained higher than the non-cancer controls in the same time period. The full set of factor loadings are available in S1 Table.

For non-insulin anti-diabetics, the proportion of adherent survivors decreased by 4 to 7 percentage points among those diagnosed with colorectal and lung cancer (Fig 1). There were no significant changes in anti-diabetic adherence among breast cancer survivors. Prostate cancer survivors experienced an increase in anti-diabetic adherence of 2 percentage points (95% Confidence Interval [CI]: 0.00–0.04) one year after diagnosis. A similar pattern was observed for adherence to statins with no significant changes among breast cancer survivors, 4 to 6 percentage point reductions among colorectal and lung cancer survivors, and a small increase of 1 percentage point (95% CI: 0.00–0.003) among prostate cancer survivors at both time points (Fig 2). The proportion of survivors adherent to anti-hypertensives increased among breast cancer survivors at both time points by 2 percentage points (95% CI: 0.02–0.03), increased by 2 percentage points (95% CI: 0.01–0.03) among colorectal cancer survivors at both time points, and increased by 4 percentage points (95% CI: 0.03–0.04) among prostate cancer survivors at both time points (Fig 3). Discontinuation rates increased among all cancer cohorts at both time points for anti-diabetics and statins but less so for anti-hypertensives (Fig 1 in S1 Fig).

Next, for each statistically significant TE, we tested for statistically significant NIE (combined across the two mediating factors) as evidence of mediation through the provider team. Only 4 of the 17 cancer/chronic condition/time combinations with significant TE had statistically significant NIE. Even in those 4 analytic cohorts, the NIE were practically very small. For

**Table 1. Provider team characteristics by cancer and time period: Mean (standard deviation).**

| Cancer | Time[d] | Factor: Number[a] | | | | Factor: Sharing | | | |
| --- | --- | --- | --- | --- | --- | --- | --- | --- | --- |
| | | Providers | | Specialists | | Degree[b] | | Shared Patient Volume[c] | |
| | | Cancer | Control | Cancer | Control | Cancer | Control | Cancer | Control |
| Breast (N = 11,831) | Diagnosis | 1.99 | 1.84 | 1.26 | 1.18 | 51.43 | 49.87 | 0.014 | 0.014 |
| | | (1.57) | (1.65) | (0.64) | (0.64) | (57.04) | (59.59) | (0.049) | (0.052) |
| | 1 year | 3.60 | 1.92 | 2.13 | 1.19 | 102.06 | 52.74 | 0.022 | 0.014 |
| | | (1.99) | (1.78) | (0.72) | (0.65) | (100.69) | (61.78) | (0.047) | (0.053) |
| | 2 years | 3.21 | 1.97 | 1.97 | 1.20 | 86.86 | 53.24 | 0.018 | 0.014 |
| | | (2.09) | (1.85) | (0.77) | (0.66) | (96.53) | (64.02) | (0.046) | (0.051) |
| Colorectal (N = 6,580) | Diagnosis | 2.60 | 1.92 | 1.60 | 1.24 | 61.90 | 53.44 | 0.014 | 0.014 |
| | | (2.11) | (1.76) | (0.79) | (0.70) | (63.10) | (61.44) | (0.045) | (0.052) |
| | 1 year | 3.34 | 1.99 | 1.89 | 1.24 | 101.05 | 55.48 | 0.021 | 0.015 |
| | | (2.48) | (1.92) | (0.79) | (0.71) | (96.51) | (63.88) | (0.047) | (0.053) |
| | 2 years | 2.99 | 2.09 | 1.75 | 1.26 | 85.95 | 56.87 | 0.018 | 0.014 |
| | | (2.45) | (2.10) | (0.80) | (0.71) | (92.37) | (67.34) | (0.048) | (0.051) |
| Lung (N = 4,105) | Diagnosis | 2.96 | 1.93 | 1.71 | 1.24 | 74.67 | 54.24 | 0.013 | 0.014 |
| | | (2.36) | (1.86) | (0.81) | (0.70) | (79.28) | (63.98) | (0.046) | (0.048) |
| | 1 year | 3.98 | 2.04 | 2.11 | 1.25 | 115.14 | 56.73 | 0.020 | 0.015 |
| | | (2.74) | (2.04) | (0.72) | (0.72) | (104.39) | (66.26) | (0.044) | (0.057) |
| | 2 years | 3.88 | 2.15 | 2.05 | 1.27 | 106.19 | 58.46 | 0.016 | 0.014 |
| | | (3.02) | (2.34) | (0.78) | (0.73) | (109.21) | (69.37) | (0.042) | (0.050) |
| Prostate (N = 11,879) | Diagnosis | 2.52 | 1.91 | 1.87 | 1.26 | 75.35 | 54.41 | 0.014 | 0.013 |
| | | (1.56) | (1.72) | (0.74) | (0.74) | (73.88) | (65.13) | (0.034) | (0.048) |
| | 1 year | 2.90 | 1.96 | 1.95 | 1.27 | 87.19 | 57.55 | 0.018 | 0.013 |
| | | (1.82) | (1.87) | (0.71) | (0.75) | (84.00) | (68.52) | (0.037) | (0.047) |
| | 2 years | 2.80 | 2.06 | 1.87 | 1.28 | 79.33 | 58.31 | 0.016 | 0.014 |
| | | (1.93) | (2.06) | (0.76) | (0.76) | (82.98) | (71.37) | (0.035) | (0.052) |

[a] Providers and Specialists represent the total number of providers and number of specialists, respectively, seen by each person in the period.

[b] Degree is the count of all providers that share patients with the patient's main provider.

[c] For each pair of providers on the patient's team, we calculated the proportion of each providers' patients who were shared with the other provider in the dyad in the previous period. Shared patient volume is the geometric mean all pairwise proportions.

[d] Six months ending at the time period indicated (relative to diagnosis date).

example, among lung cancer survivors, adherence to statins decreased by 5 percentage points (95% CI: -0.08 –-0.03); the NDE was a 4.5 percentage point decrease (95% CI: -0.07 –-. 02) and the NIE represented only 16% (95% CI: 2–30) of the TE (Fig 4). Similarly, the significant increases in adherence to anti-hypertensives among colorectal cancer survivors (at two years

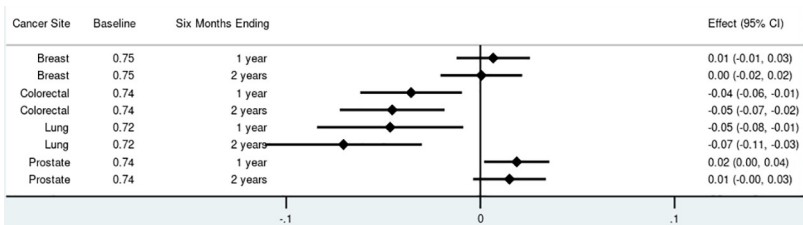

**Fig 1. Total effect of cancer on proportion adherent (proportion of days covered > 80%) for non-insulin anti-diabetics by cancer site and phase of care.** Point estimates and 95% confidence intervals.

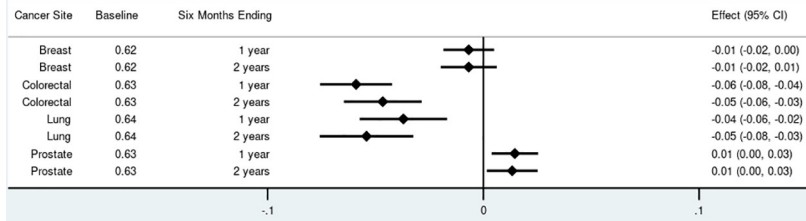

**Fig 2. Total effect of cancer on proportion adherent (proportion of days covered > 80%) for statins by cancer site and phase of care.** Point estimates and 95% confidence intervals.

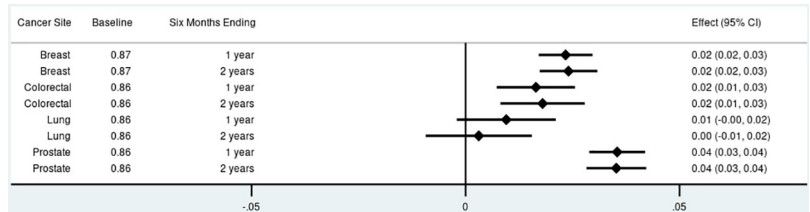

**Fig 3. Total effect of cancer on proportion adherent (proportion of days covered > 80%) for anti-hypertensives by cancer site and phase of care.** Point estimates and 95% confidence intervals.

post-diagnosis) and prostate cancer survivors were mostly due to NDE and not mediation through the provider team (Fig 5). The proportion of the TE mediated through the NIE was 18% (95% CI: 4–31) among colorectal cancer survivors, 6% (95% CI: 3–9) among prostate cancer survivors at one year and 3% (95% CI: 1–4) among prostate cancer survivors at two years.

For discontinuation, 10 of the 19 analytic cohorts with significant TE also had a statistically significant NIE (Fig 2 in S1 Fig). In each case, the NIE worked to offset the increase in

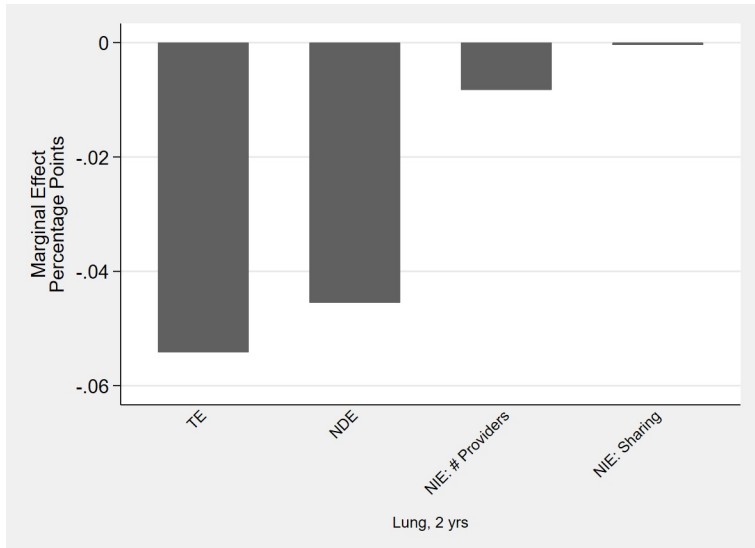

**Fig 4. Effects of cancer on proportion adherent (proportion of days covered > 80%) for statins: Total Effect (TE), Natural Direct Effect (NDE) and Natural Indirect Effect (NIE) through number of providers and sharing amongst providers.** Point estimates, in percentage point changes, for cancer sites and phases of care with statistically significant total and net indirect effect.

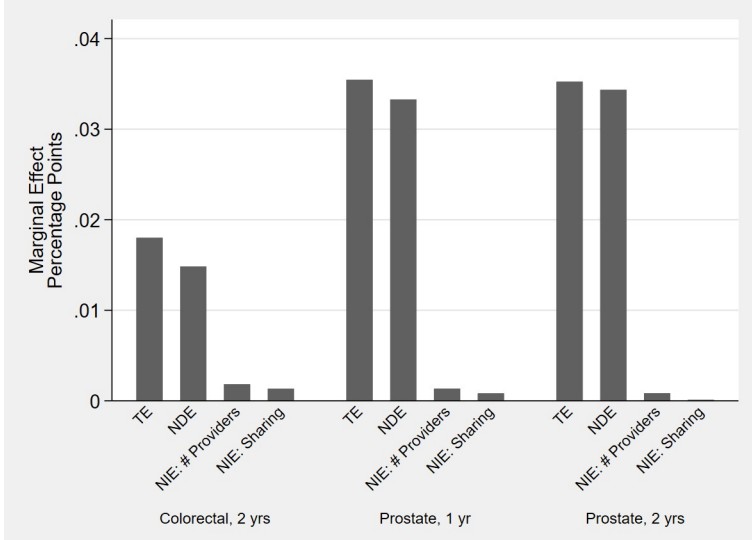

**Fig 5. Effects of cancer on proportion adherent (proportion of days covered $> 80\%$) for anti-hypertensives: Total Effect (TE), Natural Direct Effect (NDE) and Natural Indirect Effect (NIE) through number of providers and sharing amongst providers.** Point estimates, in percentage point changes, for cancer sites and phases of care with statistically significant total and net indirect effect.

discontinuation from the NDE (i.e., survivors who saw more providers/specialists and whose providers shared more patients were *less* likely to discontinue). However, the magnitude of the mediating effects (i.e., NIE) were small relative to the direct effect of cancer on discontinuation (i.e., NDE).

## Discussion

A growing body of literature demonstrates significant changes in chronic disease medication adherence associated with cancer diagnosis and treatment [7–13]. The appropriate points for intervention to improve adherence, however, remain undefined. We found that changes in the provider team structure accompanying cancer diagnosis explained only a small portion of changes in chronic condition medication adherence. Provider team structure effects were statistically insignificant in 13 of 17 analytic samples with significant changes in medication adherence. When the provider team structure effects were statistically significant, they were small in magnitude ($<$0.5 percentage points). The results for provider team structure effects were similar for discontinuation.

What do these findings mean for policy and clinical interventions? As payers move toward value-based care, providers are assuming more financial risk for holistic patient outcomes. In bundled payment arrangements like the Oncology Care Model, providers are responsible for all care delivered to cancer patients on chemotherapy. For example, inpatient admissions for uncontrolled hypertension or diabetes will hurt quality metrics and, therefore, the shared savings amounts for participating providers. Thus, these arrangements provide incentives for providers to better coordinate care to improve adherence to chronic disease medications among cancer patients.

This is the first study, to our knowledge, to disentangle the influence of care coordination from other factors that might influence chronic medication adherence in oncology. Quality improvement efforts have emphasized care coordination during transitions from oncology specialty care to primary care providers [15] and improved care coordination across a patient's

provider team [14, 16–18]. However, our results suggest that the role of increased complexity in the provider team may be outsized by other factors that can contribute to poor medication adherence for comorbid chronic conditions. This highlights a need to better understand the role of other factors that may reduce medication adherence in oncology populations.

While this paper provides evidence about complexity of the provider team, several other factors may influence adherence to chronic medications including other provider characteristics (e.g., communication, reimbursement arrangements), social determinants of health, the financial burden of cancer treatment, and physiological responses to cancer and its treatments. Social determinants of health are the conditions in the environments where people live (e.g., transportation, stable housing) and have been linked to medication adherence [42]. The financial toxicity of cancer treatment is a known constraint for many patients [43, 44], which may translate beyond cancer into the management of comorbid chronic conditions. Additionally, the chemotherapeutic toxicity of many cancer treatments (e.g., emetogenic chemotherapy) can reduce a patient's adherence to comorbid medications [45]. Furthermore, physiological responses may result in imbalances in blood pressure and blood glucose control, which can influence medication adherence [8, 46]. It is important to understand the range of factors that may influence chronic condition medication adherence as the interventions to improve each of these factors vary significantly.

This study had several strengths. First, we included survivors with one or more of several chronic conditions diagnosed with one of the four most common cancer sites [13]. Second, our estimation approach separates the effects of cancer diagnosis on medication adherence from underlying aging trends using a matched cohort of non-cancer controls. Our estimates also adjust for other factors influencing adherence that do not change over time (e.g., "healthy users," health literacy, tumor stage). Finally, we report short- (one year) and long-run (two years) effects of cancer on chronic condition medication adherence beyond initial treatment.

This study also had limitations. First, medication adherence was evaluated using dispensed prescriptions and we cannot assume that all filled medications were consumed. Second, this study used data from 2008–2014 and was restricted to adults age 66 and older with continuous Medicare fee-for-service and Part D coverage with non-metastatic cancer who also survived two years following their cancer diagnosis. As such, our findings may not be generalizable to more current patients, those with Medicare Advantage or without prescription drug or other healthcare insurance, the population 65 years and under, patients with metastatic disease, or patients with a short life expectancy. Furthermore, we were not able to assess the effect of care coordination on mortality. Third, we cannot determine whether the observed changes in chronic condition medication adherence were clinically appropriate. Fourth, several mechanisms exist for the NDE of cancer (e.g., competing long-term cancer therapies like endocrine therapy) and more research is needed to unpack these mechanisms. Finally, causal interpretation of our results depends on the causal model and accompanying identification assumptions being correct.

This study found that changes in medication adherence due to cancer diagnosis differed across cancer sites and chronic conditions. The largest decreases in chronic condition medication adherence occurred for anti-diabetics and statins among colorectal and lung cancer survivors, while adherence to anti-hypertensives increased among breast, colorectal and prostate cancer survivors. The decreases in adherence among colorectal and lung cancer patients are consistent with a hypothesis that patients diagnosed with more deadly cancer have decreased incentive to manage chronic conditions, perhaps because preventive medication has a long lag-time for benefit [47] or because more complex treatments (e.g., surgical resection and adjuvant chemotherapy) change the need or benefits of continued use of chronic disease medications [13]. Conversely, patients diagnosed with cancers with high survival probabilities (e.g.,

breast and prostate) are motivated to improve their (secondary) prevention efforts, including medication adherence. The positive effects of cancer diagnosis on antihypertensive medication adherence may be explained by the fact that blood pressure monitoring is routine during cancer care visits, which provides ample opportunity for providers to promote the importance of antihypertensive adherence. Alternatively, monitoring of lipid and blood sugar levels is less routine, which might partially explain the decreases in adherence to these medications among cancer patients. Of course, other explanations are also plausible.

For all cancer sites and chronic conditions, cancer diagnosis led to increased number of providers, specialists and patient sharing among the provider team. However, the increased complexity in the provider team structure associated with cancer diagnosis did not lead to meaningful changes in medication adherence for chronic conditions. These results suggest that policies and interventions aimed at improving chronic condition medication adherence need to be targeted based on the type of cancer and chronic condition and can focus on systemic and patient factors that are present across provider teams for greater effect.

## Supporting information

**S1 Appendix. Details of causal model.**
(DOCX)

**S1 Fig. Results for discontinuation outcome.**
(DOCX)

**S1 Table. Factor weights by cancer site and chronic condition.**
(DOCX)

## Acknowledgments

This study used the linked SEER-Medicare database. The interpretation and reporting of these data are the sole responsibility of the authors. The collection of cancer incidence data used in this study was supported by the California Department of Public Health as part of the statewide cancer reporting program mandated by California Health and Safety Code Section 103885; the National Cancer Institute's Surveillance, Epidemiology and End Results Program under contract HHSN261201000140C awarded to the Cancer Prevention Institute of California, contract HHSN261201000035C awarded to the University of Southern California, and contract HHSN261201000034C awarded to the Public Health Institute; and the Centers for Disease Control and Prevention's National Program of Cancer Registries, under agreement # U58DP003862-01 awarded to the California Department of Public Health. The ideas and opinions expressed herein are those of the author(s) and endorsement by the State of California Department of Public Health, the National Cancer Institute, and the Centers for Disease Control and Prevention or their Contractors and Subcontractors is not intended nor should be inferred. The authors acknowledge the efforts of the National Cancer Institute; the Office of Research, Development and Information, CMS; Information Management Services (IMS), Inc.; and the Surveillance, Epidemiology, and End Results (SEER) Program tumor registries in the creation of the SEER-Medicare database.

## Author Contributions

**Conceptualization:** Justin G. Trogdon, Benjamin Y. Urick, Katherine E. Reeder-Hayes, Joel F. Farley, Stephanie B. Wheeler, Jennifer L. Lund.

**Data curation:** Justin G. Trogdon, Krutika Amin, Parul Gupta, Jennifer L. Lund.

**Formal analysis:** Justin G. Trogdon, Krutika Amin, Parul Gupta.

**Funding acquisition:** Justin G. Trogdon, Jennifer L. Lund.

**Investigation:** Justin G. Trogdon, Katherine E. Reeder-Hayes, Joel F. Farley, Stephanie B. Wheeler, Lisa Spees, Jennifer L. Lund.

**Methodology:** Justin G. Trogdon, Benjamin Y. Urick, Stephanie B. Wheeler, Jennifer L. Lund.

**Project administration:** Justin G. Trogdon.

**Writing – original draft:** Justin G. Trogdon, Jennifer L. Lund.

**Writing – review & editing:** Krutika Amin, Parul Gupta, Benjamin Y. Urick, Katherine E. Reeder-Hayes, Joel F. Farley, Stephanie B. Wheeler, Lisa Spees.

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
