## [Decision Letter · Decision Letter 0]

14 Jun 2021

PONE-D-21-09579

Providers’ Mediating Role for Medication Adherence among Cancer Survivors

PLOS ONE

Dear Dr. Trogdon,

Thank you for submitting your manuscript to PLOS ONE. After careful consideration, we feel that it has merit but does not fully meet PLOS ONE’s publication criteria as it currently stands. Therefore, we invite you to submit a revised version of the manuscript that addresses the points raised during the review process.

Please pay special attention to the concerns raised by reviewer 2 on the discussion

We look forward to receiving your revised manuscript.

Kind regards,

Ilana Graetz

Academic Editor

PLOS ONE

Journal Requirements:

3) In the ethics statement in the manuscript and in the online submission form, please provide additional information about the patient records used in your retrospective study, including the date range (month and year) during which patients' medical records were accessed.

4)  We note that you have indicated that data from this study are available upon request. PLOS only allows data to be available upon request if there are legal or ethical restrictions on sharing data publicly. For information on unacceptable data access restrictions, please see http://journals.plos.org/plosone/s/data-availability#loc-unacceptable-data-access-restrictions.

5)  We note that the grant information you provided in the ‘Funding Information’ and ‘Financial Disclosure’ sections do not match.

6) Thank you for stating the following in the Competing Interests section:

[Dr. Lund’s spouse is a full-time, paid employee of GlaxoSmithKline who also holds

stock in the amount of approximately $42,000. Dr. Lund also receives unrelated grant

funding paid to her institution from AbbVie. Dr. Wheeler receives unrelated grant

funding paid to her institution from Pfizer. All other co-authors have no potential

competing interests to report.].

7) Please include captions for your Supporting Information files at the end of your manuscript, and update any in-text citations to match accordingly. Please see our Supporting Information guidelines for more information: http://journals.plos.org/plosone/s/supporting-information.

Reviewers' comments:

Reviewer's Responses to Questions

**Comments to the Author**

1. Is the manuscript technically sound, and do the data support the conclusions?

Reviewer #1: Yes

Reviewer #2: Partly

2. Has the statistical analysis been performed appropriately and rigorously? 

Reviewer #1: Yes

Reviewer #2: Yes

3. Have the authors made all data underlying the findings in their manuscript fully available?

Reviewer #1: Yes

Reviewer #2: Yes

4. Is the manuscript presented in an intelligible fashion and written in standard English?

Reviewer #1: Yes

Reviewer #2: Yes

5. Review Comments to the Author

Reviewer #1: This is a mediation analysis of the provider team’s role in changes to chronic condition medication adherence among cancer survivors. The study topic is of interest from both clinical and policy perspective. Further, the study has a reasonable methodology, although has limitations in terms of using SEER-Medicare data and relying on pharmacy refills as a measure of adherence. There are few minor comments regarding this study:

1- It is understandable that the authors have focused on 4 most common cancers, but it is not clear why in each set of cancers they have limited their population. For example for breast cancer, why stage IV was not included? or in lung cancer, why small cell lung was not included. If there is concern about heterogeneity in management, similar heterogeneity can be expected with colorectal cancer stages as well.

2- One of the main limitation of the study is that the data is from a decade ago, and might not be generalizable to current practices.

3- In assigning provider team structure, there has been no mention of mid levels who commonly manage patients as primary care. Why mid levels were excluded from the study?

4- It is not clear why patient sharing with other providers, or count of all providers sharing patients (degree) is increasing among cancer patients vs. controls (in Table 1). Why a single patient cancer diagnosis impacting the patient volume shared?

5- Authors may want to expand on why they think there is differences in adherence based on the chronic condition (why adherence to anti hypertensives higher) or type of cancer. Also there are controversial publications in this area; for example some other papers have shown breast cancer patients will have decreased adherence to chronic medications (as opposed to this study).

6- In supplemental appendix, there is an error displaying Fig 2A-C, and the images could not be visualized.

Reviewer #2: The reviewer has high enthusiasm for this well designed and well written analysis assessing the role of the complexity of the provider team on adherence to chronic disease medications for cancer survivors. This study includes many strengths, including looking at several different cancers and using robust methods with several novel provider complexity metrics. It also offers some good news: that our healthcare system can accommodate increasing provider complexity without risking patient adherence outcomes. These comments and questions are offered in the spirit of improving the clarity of the methods and carefulness of the language in the discussion.

Methods

At first I was concerned about the exclusion about insurance, as those experience insurance losses may be most vulnerable to adherence challenges. At the same time, I believe this was a strong analytic choice, as it’s important to show that insurance changes were not the cause of non-adherence, and allows the team to further isolate the role of provider complexity.

Was there censorship due to death? That is to say, that those with poorly coordinated care might have shorter survival. I think not, given that the limitations suggest that only those surviving two years are excluded. I think this is a point worth extracting out a bit: that it’s possible that provider complexity may have been important to survival overall, but, by design, that was not assessed in this analysis.

I’m not familiar with the term “demeaning”. Is this similar to group mean centering or grand mean centering?

It seems that this analysis has time nested in patients nested in providers. Were hierarchical models used? If no hierarchical models were used, how would you account for shared variance due to clustering of patients within providers?

Some of the cancers include long-term therapies that might also compete with comorbidity management. For example, endocrine therapy for breast cancer may be last up to 5 years, and include both oral and intravenous components. Were those medications also accounted for in some way?

Results

The mediation model adds value, but I also wonder if there are moderating effects. Given the results in the figures showing differences by tumor site, have you explored whether the interaction of provider complexity with tumor site might lead to differences in adherence?

Discussion

Some of the language in the discussion extends beyond what the results of this study suggest, especially with the language about the need to focus on patient-level factors. Currently, it reads as if patient self-efficacy is the culprit for adherence challenges, as if the problem is that patients just aren’t motivated enough to be adherent. It comes across as victim-blaming and as if increasing a sense of personal responsibility will fix adherence issues. Current interventions for medication adherence for cancer survivors that focus on patient-level factors suggest otherwise1-5. If we can reasonably assume that patients do care about their health to some degree, there are likely competing priorities that may be wholly outside of health care that patients are busy addressing and influences their adherence. While this paper providers strong evidence about complexity of the provider team, there may be other provider-level factors that are still important that are not explored here. It is premature to suggest we should switch to focusing on patient factors. I recommend this language on p14 be revised substantially.

One of the values of this paper is that it gives us a start on the evidence around the influence of providers: but also leaves more room to explore. For example, might provider complexity influence disparities in adherence? Might there be rural-urban differences to consider or issues of geography? Rather than saying we should immediately shift to patient-level factors, it would be worthwhile to point to other provider-level issues to explore.

References

1. Neven P, Markopoulos C, Tanner M, et al. The impact of educational materials on compliance and persistence rates with adjuvant aromatase inhibitor treatment: First-year results from the Compliance of ARomatase Inhibitors AssessmenT In Daily practice through Educational approach (CARIATIDE) study. The Breast. 2014;23(4):393-399.

2. Ziller V, Kyvernitakis I, Knöll D, Storch A, Hars O, Hadji P. Influence of a patient information program on adherence and persistence with an aromatase inhibitor in breast cancer treatment-the COMPAS study. BMC cancer. 2013;13(1):407.

3. Hadji P, Blettner M, Harbeck N, et al. The Patient's Anastrozole Compliance to Therapy (PACT) Program: a randomized, in-practice study on the impact of a standardized information program on persistence and compliance to adjuvant endocrine therapy in postmenopausal women with early breast cancer. Annals of oncology. 2013;24(6):1505-1512.

4. Lambert LK, Balneaves LG, Howard AF, Gotay CC. Patient-reported factors associated with adherence to adjuvant endocrine therapy after breast cancer: an integrative review. Breast cancer research and treatment. 2018;167(3):615-633.

5. Greer JA, Amoyal N, Nisotel L, et al. A systematic review of adherence to oral antineoplastic therapies. The oncologist. 2016;21(3):354-376.

6. PLOS authors have the option to publish the peer review history of their article (what does this mean?). If published, this will include your full peer review and any attached files.

Reviewer #1: No

Reviewer #2: No

---

## [Author Response · Author response to Decision Letter 0]

4 Oct 2021

Thank you for the comments. We provide a detailed response to each comment in the attached Response to Reviewers (PLOS_One_response_reviewers_final_v2.docx).

---

## [Editor Report · Decision Letter 1]

9 Nov 2021

Providers’ mediating role for medication adherence among cancer survivors

PONE-D-21-09579R1

Dear Dr. Trogdon,

We’re pleased to inform you that your manuscript has been judged scientifically suitable for publication and will be formally accepted for publication once it meets all outstanding technical requirements.

Kind regards,

Ilana Graetz

Academic Editor

PLOS ONE

---

## [Editor Report · Acceptance letter]

17 Nov 2021

PONE-D-21-09579R1 

Providers’ mediating role for medication adherence among cancer survivors 

Dear Dr. Trogdon:

I'm pleased to inform you that your manuscript has been deemed suitable for publication in PLOS ONE. Congratulations! Your manuscript is now with our production department. 

Kind regards, 

on behalf of

Dr. Ilana Graetz 

Academic Editor

PLOS ONE